

# ncRNA2MetS: a manually curated database for non-coding RNAs associated with metabolic syndrome

Dengju Yao[1,2,3], Xiaojuan Zhan[4,5], Xiaorong Zhan[6], Chee Keong Kwoh[2] and Yuezhongyi Sun[1,5]

[1] School of Software and Microelectronics, Harbin University of Science and Technology, Harbin, Heilongjiang, China
[2] School of Computer Science and Engineering, Nanyang Technological University, Singapore, Singapore
[3] College of Bioinformatics Science and Technology, Harbin Medical University, Harbin, Heilongjiang, China
[4] College of Computer Science and Technology, Heilongjiang Institute of Technology, Harbin, Heilongjiang, China
[5] School of Computer Science and Technology, Harbin University of Science and Technology, Harbin, Heilongjiang, China
[6] Department of Endocrinology and Metabolism, the First Affiliated Hospital of Harbin Medical University, Harbin, Heilongjiang, China

Corresponding author
Dengju Yao, ydkvictory@hrbust.edu.cn

## ABSTRACT

Metabolic syndrome is a cluster of the most dangerous heart attack risk factors (diabetes and raised fasting plasma glucose, abdominal obesity, high cholesterol and high blood pressure), and has become a major global threat to human health. A number of studies have demonstrated that hundreds of non-coding RNAs, including miRNAs and lncRNAs, are involved in metabolic syndrome-related diseases such as obesity, type 2 diabetes mellitus, hypertension, etc. However, these research results are distributed in a large number of literature, which is not conducive to analysis and use. There is an urgent need to integrate these relationship data between metabolic syndrome and non-coding RNA into a specialized database. To address this need, we developed a metabolic syndrome-associated non-coding RNA database (ncRNA2MetS) to curate the associations between metabolic syndrome and non-coding RNA. Currently, ncRNA2MetS contains 1,068 associations between five metabolic syndrome traits and 627 non-coding RNAs (543 miRNAs and 84 lncRNAs) in four species. Each record in ncRNA2MetS database represents a pair of disease-miRNA (lncRNA) association consisting of non-coding RNA category, miRNA (lncRNA) name, name of metabolic syndrome trait, expressive patterns of non-coding RNA, method for validation, specie involved, a brief introduction to the association, the article referenced, etc. We also developed a user-friendly website so that users can easily access and download all data. In short, ncRNA2MetS is a complete and high-quality data resource for exploring the role of non-coding RNA in the pathogenesis of metabolic syndrome and seeking new treatment options. The website is freely available at http://www.biomed-bigdata.com:50020/index.html

## INTRODUCTION

Metabolic syndrome (MetS) is a cluster of the most dangerous heart attack risk factors: diabetes and raised fasting plasma glucose, abdominal obesity, high cholesterol and high blood pressure (*Alberti, Zimmet & Shaw, 2005*). It is estimated that around 20–25% of the world's adult population have metabolic syndrome, making them three times more likely to have, and twice as likely to die from, a heart attack or stroke when compared to people without the syndrome (*International Diabetes Federation, 2006*). Metabolic syndrome has become a major threat to human health around the world. However, to date, the pathogenesis of metabolic syndrome continues to challenge experts. In recent years, a growing number of studies have suggested that many non-coding RNAs (ncRNAs), including small non-coding RNAs, particularly microRNAs (miRNAs), and long non-coding RNAs (lncRNAs), may be involved in metabolic syndrome-related diseases such as obesity, type 2 diabetes mellitus, hypertension etc. (*Stoll et al., 2018*; *La Sala et al., 2018*; *Esguerra et al., 2018*; *Cui et al., 2018*; *Lorente-Cebrián et al., 2019*). Dysregulation of some miRNAs and lncRNAs disrupts the gene regulatory network, leading to metabolic syndrome and other related diseases. MiRNAs are ~22 nt non-coding small RNAs that negatively regulate gene expression at the post-transcriptional level (*Bartel, 2004*). Extensive research suggests that many miRNAs, such as miR-9 (*Hu et al., 2018*), miR-20b-5p (*Katayama et al., 2019*; *Gentile et al., 2019*), miR-802 (*Kornfeld et al., 2013*), let-7f (*Gentile et al., 2019*), miR-33 (*Rayner et al., 2010*), miR-375 (*Sedgeman et al., 2019*), and others are involved in glucose homeostasis, diabetes mellitus, abdominal obesity and cholesterol metabolism. Furthermore, some lncRNAs, a novel class of long non-coding RNA larger than 200 nt, have been reported to be involved in the pathogenesis of type 2 diabetes mellitus and metabolic syndrome (*Singer & Sussel, 2018*; *Losko, Kotlinowski & Jura, 2016*; *Wang et al., 2018*).

Due to the important effects of metabolic syndrome on human health, it is urgent to develop a database dedicated to various biomarkers associated with metabolic syndrome, such as genes, miRNAs and lncRNAs. In recent years, several data resources and tools have been developed for storing metabolic disease-associated biomolecules, such as T-HOD, metabolicMine, PathCaseMAW, HMA and BioM2MetDisease. T-HOD (*Dai et al., 2013*) is a literature-based candidate gene database currently containing 837, 835 and 821 candidate genes for hypertension, obesity and diabetes, respectively. metabolicMine (*Lyne et al., 2013*) is a data warehouse with a specific focus on the genomics, genetics and proteomics of common metabolic diseases. PathCaseMAW (*Cicek et al., 2013*) provides a database-enabled framework and web-based computational tools for browsing, querying, analyzing and visualizing stored metabolic networks. HMA (*Pornputtapong, Nookaew & Nielsen, 2015*) is a human metabolic atlas website which provides information about human metabolism. These four software resources described above have provided important support for the study of the pathogenesis of metabolic diseases. However, they do not contain non-coding RNA information related to metabolic diseases. BioM2MetDisease (*Xu et al., 2017*) is a manually curated database containing 2,681 entries of associations between 1,147 biomolecules and 78 metabolic diseases. Though it is a very useful tool for
studying metabolic diseases, BioM2MetDisease is not a database dedicated to metabolic syndrome. It contains miRNAs associated with 78 metabolic diseases but does not include hypertension and hypo-HDL cholesterolemia, which are two important traits of metabolic syndrome. In addition, it does not contain lncRNAs associated with metabolic syndrome and miRNAs from the last two years.

In addition to the five data resources described above, there are other resources and tools for studying human diseases. miR2Disease (*Jiang et al., 2009*) is a manually curated database which contains 1,939 curated relationships between 299 human miRNAs and 94 human diseases. phenomiR (*Ruepp, Kowarsch & Theis, 2012*) provides miRNA and target relations from these studies on the association of dysregulated miRNAs and diseases. HMDB (v3.0) (*Wishart et al., 2013*) is a resource dedicated to the human metabolome, which includes more than 40,000 annotated metabolite entries. HMDD (v3.0) (*Huang et al., 2018*) manually collects a significant number of miRNA-disease association entries. All these databases have provided valuable tools for exploring the roles of these biomolecules in human diseases, but they are not designed specifically for metabolic syndrome. When faced with so many data resource options, it is difficult to find the desired data for doctors who focus on metabolic syndrome. In addition, more and more non-coding RNAs associated with metabolic syndrome have been discovered in the last two years (*Saeedi Borujeni et al., 2019*; *Zhang et al., 2019*; *Zhang, Li & Reilly, 2019*; *Smieszek et al., 2019*; *Li et al., 2019*).

So far, there is still no non-coding RNA database dedicated to metabolic syndrome. There is an urgent need for a specialized data resource containing all the latest non-coding RNAs associated with metabolic syndrome. To meet this demand, we have developed the ncRNA2MetS database which contains the latest and most complete MetS-miRNA (lncRNA) associations validated by various biological experiments. We carefully reviewed 571 articles about relationship between various metabolic syndrome traits and non-coding RNA and gained 1,068 associations between five metabolic syndrome traits and 627 non-coding RNAs (543 miRNAs and 84 lncRNAs) in four species. We hope that this database can help doctors specializing in metabolic syndrome to explore the pathogenesis and treatments of metabolic syndrome.

## MATERIALS & METHODS
### Data collection from literature in PubMed
According to the International Diabetes Federation (IDF) definition, a person with metabolic syndrome must have central obesity (defined as waist circumference with ethnicity-specific values) plus any two of the following four factors: (1) raised triglycerides, or specific treatment for this lipid abnormality; (2) reduced HDL cholesterol, or specific treatment for this lipid abnormality; (3) raised blood pressure, or treatment of previously diagnosed hypertension; (4) raised fasting plasma glucose, or previously diagnosed type 2 diabetes (*Alberti, Zimmet & Shaw, 2005*; *International Diabetes Federation, 2006*). Therefore, we divided the risk factors of metabolic syndrome into five traits: central obesity, type 2 diabetes mellitus, hypertension, hypertriglyceridemia and hypo-HDL cholesterolemia.

Referring to Xu's method (*Xu et al., 2017*), we manually collected and curated MetS-miRNA (lncRNA) associations from related articles in the PubMed database. First, we used 'non-coding RNA', 'ncRNA', 'microRNA', 'miRNA', 'long non-coding RNA', 'lncRNA' and each metabolic syndrome trait as search terms to search the PubMed database by Title/Abstract retrieval method. As a result, we gained more than 3,000 related articles published since 2007. We filtered out a large number of irrelevant articles or reviews by reading abstracts and finally selected 571 articles that were really focused on association between metabolic syndrome trait and miRNA (lncRNA). Then, we manually extracted MetS-miRNA (lncRNA) associations by reading these selected articles in detail. In the process of extracting these associations, the detailed information about MetS-miRNA (lncRNA) association were collected, including non-coding RNA category, miRNA (lncRNA) name, name of metabolic syndrome trait, ICD-11 classification and DO (Disease Ontology) identifier for metabolic syndrome trait, method for validation (e.g., RNA-seq, luciferase report assays, gene knock-out), detected tissue (e.g., serum, adipose tissue, liver), expressive patterns (e.g., up-regulation, down-regulation, differential expression), name of the gene regulated by miRNA (lncRNA), species involved (e.g., homo sapiens, mus musculus, rattus norvegicus), referenced article (PubMed ID, title, year of publication) and a brief introduction to this association in the referenced article. Following previous research rules (*Jiang et al., 2009*; *Xu et al., 2017*; *Huang et al., 2018*), we only collected MetS-miRNA (lncRNA) associations validated by various biological experiments in this process. At the same time, in order to ensure the authenticity and reliability of the extracted information, each MetS-miRNA (lncRNA) association was confirmed by at least two scholars. Finally, in order to ensure consistency with other data resources, we standardized the names of miRNAs, lncRNAs and metabolic syndrome traits. We provided miRBase (*Kozomara, Birgaoanu & Griffiths-Jones, 2019*) identifier for miRNAs, NONCODE (*Fang et al., 2017*) identifier for lncRNAs, ICD-11 classification and DO identifier for metabolic syndrome trait. The process of constructing the ncRNA2MetS database is shown in Fig. 1.

### Database and website development

In order to facilitate users to access and use the data in the ncRNA2MetS database, we developed a user-friendly website providing data browsing, searching and downloading function. The website was implemented in Java programing language, and all data were stored in MySQL database. The website is freely available at http://www.biomed-bigdata.com:50020/index.html.

## RESULTS

### Database contents

By April 2019, we gained 3,699 potential articles from PubMed using ''Title/Abstract'' searching. After manual screening according to the relevance of the research contents, 571 articles were selected for reading in detail, and 1,068 MetS-miRNA (lncRNA) associations were identified finally. To describe MetS-miRNA (lncRNA) association in more detail, each record about MetS-miRNA (lncRNA) association in ncRNA2MetS database consists of non-coding RNA category, miRNA (lncRNA) name, miRBase identifier for miRNA,

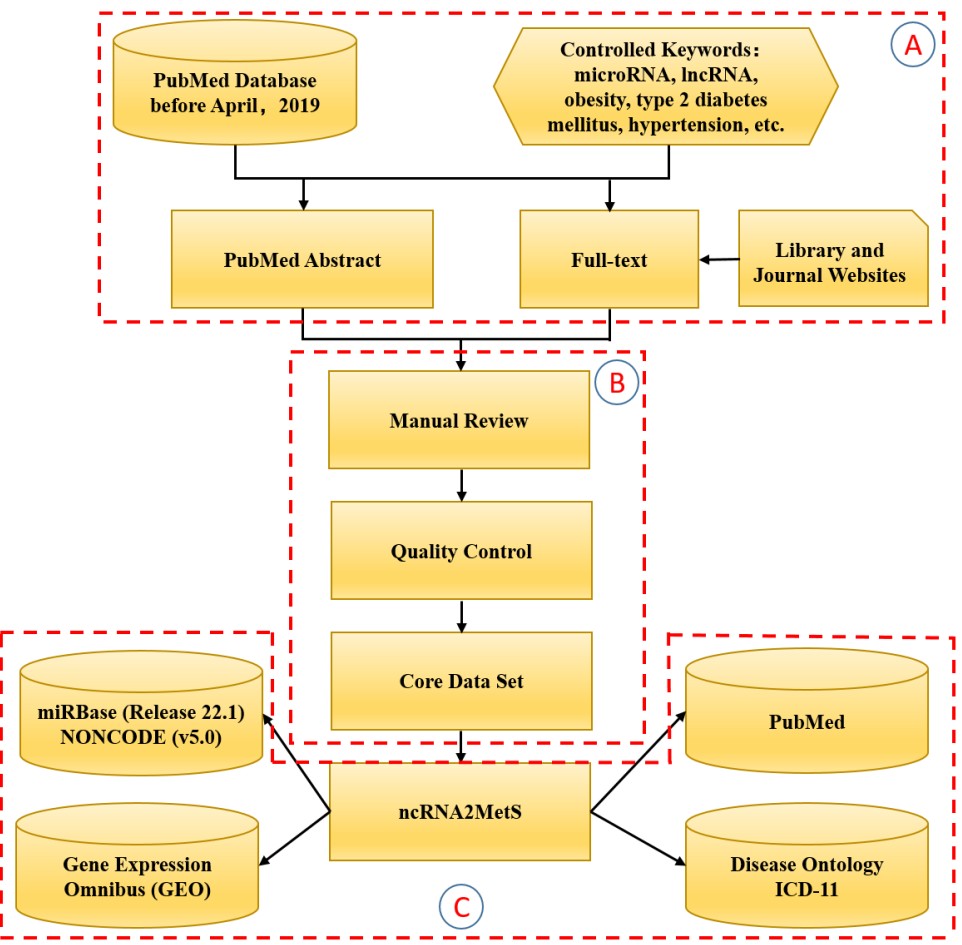

**Figure 1** **The flowchart of the ncRNA2MetS database design.** The whole process is divided into three stages: (A) Literature retrieval; (B) Data extraction; (C) Database and website development.

NONCODE identifier for lncRNA, name of metabolic syndrome trait, ICD-11 classification and DO identifier for metabolic syndrome trait, method for validation, detected tissue, expressive patterns, name of the gene regulated by miRNA (lncRNA), species involved, information of cited articles and a brief introduction to the associations in this referenced article. (see 'Materials and Methods'). Currently, the ncRNA2MetS database contains 1,068 associations between five metabolic syndrome traits (central obesity, type 2 diabetes mellitus, hypertension, hyperlipidemia and hypo-HDL cholesterolemia) and 627 non-coding RNAs (543 miRNAs and 84 lncRNAs) in four species (homo sapiens, mus musculus, rattus norvegicus and Sus scrofa). Among the 1,068 associations, the number of miRNAs related to central obesity, type 2 diabetes mellitus, hypertension, hyperlipidemia and hypo-HDL cholesterolemia are 288, 207, 96, 50 and 41, respectively. In addition, the number of miRNAs reported to be related to metabolic syndrome is 36 (Fig. 2A). The number of lncRNAs related to central obesity, type 2 diabetes mellitus, hypertension,

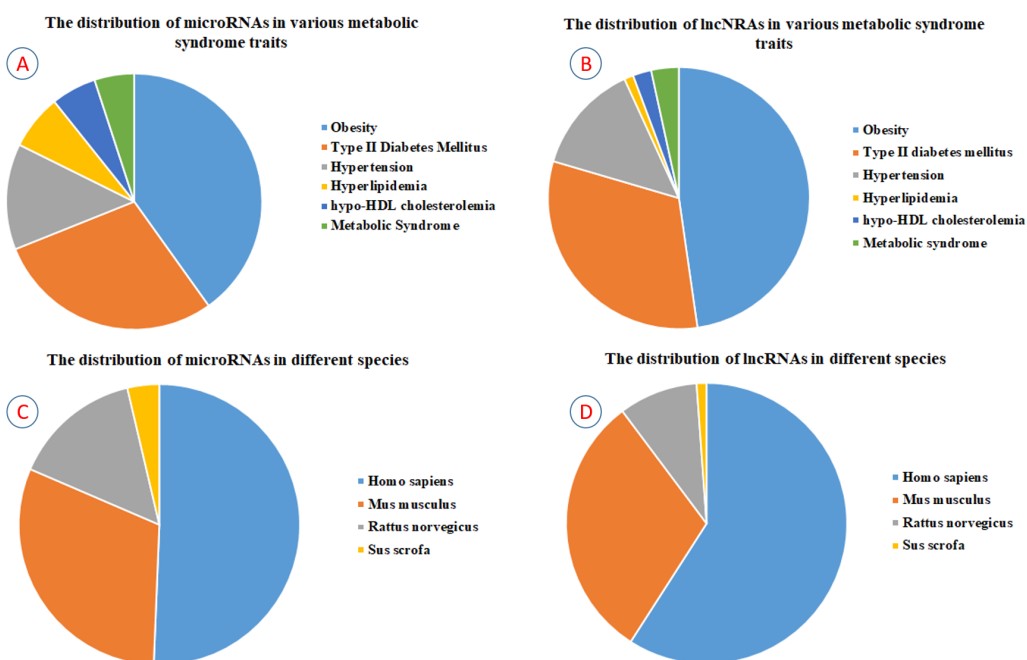

**Figure 2** **The statistics of ncRNAs contained in the ncRNA2MetS database.** (A) The distribution of miRNAs in various metabolic syndrome traits. (B) The distribution of lncRNAs in various metabolic syndrome traits. (C) The distribution of miRNAs in different species. (D) The distribution of lncRNAs in different species.

hyperlipidemia, hypo-HDL cholesterolemia and metabolic syndrome are 42, 28, 12, one, two and three, respectively (Fig. 2B).

## Database interface

ncRNA2MetS database implements a user-friendly website interface through which users can access all data in ncRNA2MetS conveniently and easily. The website consists of six parts, namely HOME, BROWSE, SEARCH, DOWNLOAD, SUBMIT and HELP. The 'HOME' page shows a brief introduction about metabolic syndrome, miRNA and lncRNA while 'BROWSE' page (Fig. 3A) and 'SEARCH' page (Fig. 3B) provide data query. On the 'BROWSE' page, users can click a specific miRNA, lncRNA or metabolic syndrome trait to browse the MetS-ncRNA associations. Then, the website will return all MetS-ncRNA associations that meet the query criteria. If too many association entries are returned, users can specify ncRNA category, species or validation method to screen for required entries. For example, if users specify 'homo sapiens' as species, the website will return all associations related to 'homo sapiens' (Fig. 3C).

The 'SEARCH' page provides users with faster and more accurate query method, which support 'Accurate Search' and 'Fuzzy Search'. For 'Accurate Search', users can input an accurate miRNA (lncRNA) name or metabolic syndrome trait name, or both, then click 'Search' button to query the required associations. On the contrary, for 'Fuzzy Search', users only need to input partial names of a miRNA (lncRNA) or metabolic syndrome

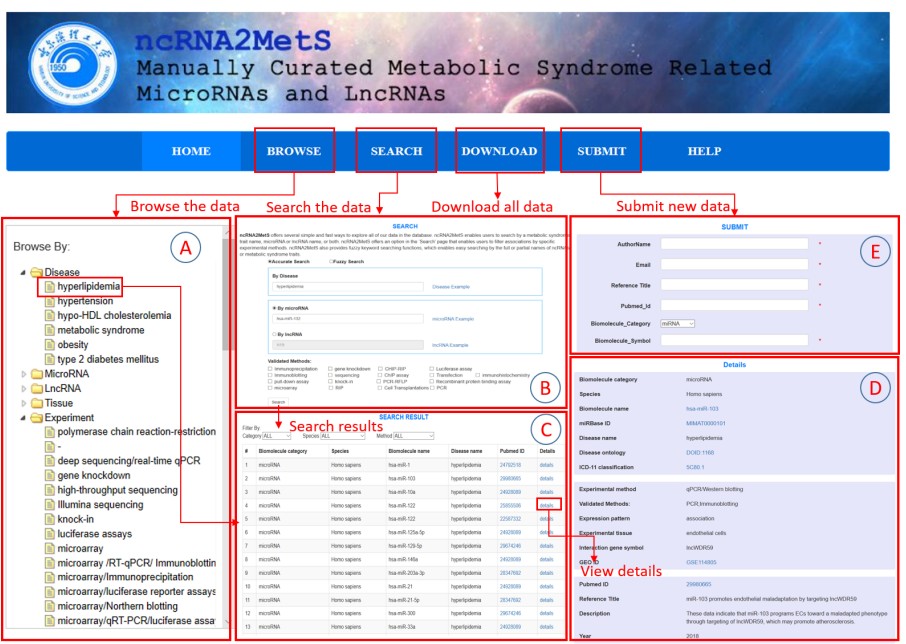

**Figure 3** **The schematic workflow of the ncRNA2MetS database.** (A) Browse the data. (B) Search the data. (C) Browse the query results. (D) Browse the detail information about a specific MetS-ncRNA association. (E) Submit a new MetS-ncRNA association to the ncRNA2MetS database.

trait to query the required relationships. It should be noted that query keywords are case-insensitive. In addition, users can specify 'Validation Methods' to narrow query scope. For example, if the user specifies 'qRT-PCR' as the validation method, the website will return all associations that have been validated by qRT-PCT. Similar to 'BROWSE', in the search result page, users can also filter MetS-ncRNA associations by selecting specified ncRNA category, species or validation method. Finally, users can browse the detailed information about a specific MetS-ncRNA association by clicking a hyperlink to the 'Details' page (Fig. 3D).

In addition to freely querying MetS-ncRNA associations stored in ncRNA2MetS, users can also submit novel associations validated by their own experiments. They can do this on the 'SUBMIT' page (Fig. 3E) and must provide detailed information about the new association. Our committee will regularly review new submissions. Once the submitted association is confirmed, it will be added into ncRNA2MetS. Furthermore, users can freely and easily download all MetS-ncRNA associations in the 'DOWNLOAD' page. Finally, if users encounter any difficulties or problems in using the ncRNA2MetS, they can find help information on 'HELP' page or contact us via e-mail.

## Examples of using ncRNA2MetS

In this section, we will use examples to show you how to use the ncRNA2MetS database. First, users can input 'miR-155-5p' as the miRNA name on the 'SEARCH' page and then click the 'Search' button. A result page will be returned and will display all records about miR-155-5p, including those of different species and various metabolic syndrome

traits. On the result page, one can easily notice that miR-155-5p is related to all five metabolic syndrome traits, which implies that it has a very important impact on metabolic syndrome. In fact, miR-155-5p has been reported to be a risk factor of metabolic syndrome. ncRNA2MetS also supports searching by the name of metabolic syndrome trait, and this can facilitate the study of pathogenesis for a specific metabolic syndrome trait. For example, users can input 'obesity' as the name of metabolic syndrome trait on the 'SEARCH' page and will find that a number of miRNAs and lncRNAs such as miR-21, miR-155-5p and Paral1 showed abnormal expression in human obesity. The introductions to these associations in ncRNA2MetS show that a reduced level of miR-21 might be associated with obesity and its related metabolic traits such as hyperinsulinemia (*Ghorbani et al., 2018*); Obese subjects have increased expressions of miR-155-5p and miR-122, two miRNAs related to inflammation and iron metabolism, respectively, at both the systemic and sperm levels (*López et al., 2018*); Furthermore, a novel component of the adipogenic transcriptional regulatory network defining the lncRNA Paral1 is identified as an obesity-sensitive regulator of adipocyte differentiation and function (*Firmin et al., 2017*).

As a feature, ncRNA2MetS also supports querying lncRNAs associated with metabolic syndrome. For example, by inputting 'H19', the result page will show all records of relationships between 'H19' and various metabolic syndrome traits including obesity, type 2 diabetes mellitus and hypo-HDL cholesterolemia in three species. The functional descriptions in ncRNA2MetS show that imprinted lncRNA H19 increases upon cold-activation and decreases in obesity in BAT (*Schmidt et al., 2018*); Related studies reveal a previously undescribed double-negative feedback loop between sponge lncRNA and target miRNA that contributes to glucose regulation in muscle cells (*Gao et al., 2014*); A H19-miR130b pathway regulating lipid metabolism and inflammation response in ox-LDL-treated Raw264.7 cells provides new targets for atherosclerosis treatment (*Han et al., 2018*). Overall, ncRNA2MetS can be used as a high-quality and most complete data resource for studying the roles of miRNAs and lncRNAs involved in metabolic syndrome.

## Database analysis

Currently, ncRNA2MetS provides almost all the research results related to the association between metabolic syndrome and non-coding RNA. Comprehensive analysis of the data in ncRNA2MetS can help people better explore the relationship between metabolic syndrome and non-coding RNA. For this purpose, a relational network between metabolic syndrome traits and ncRNAs (miRNAs and lncRNAs) is constructed using Cytoscape software (Fig. 4). In the MetS-ncRNA association network, nodes represent metabolic syndrome traits and ncRNAs, and edges represent the relationships between them. For miRNAs and lncRNAs, degree represents the number of associated metabolic syndrome traits, and also indicates their importance for researching the pathogenesis and treatment of metabolic syndrome. Figure 4 shows that the degrees of miR-122, miR-155-5p and miR-146a-5p (green node) were largest among the numerous miRNAs, which are related to all five metabolic syndrome traits (obesity, type 2 diabetes mellitus, hypertension, hyperlipidemia and hypo-HDL cholesterolemia), or have been reportedly involved in metabolic syndrome. This result implies that these three miRNAs play an important role in the study of metabolic

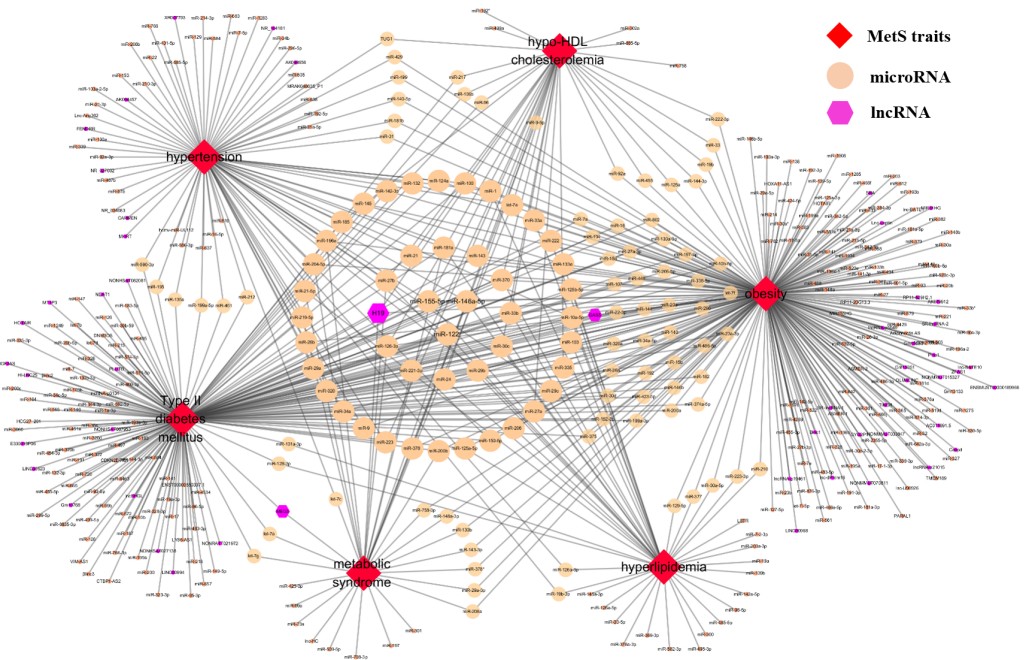

**Figure 4  The MetS-ncRNA association network.** Nodes correspond to ncRNAs (miRNAs and lncRNAs) and metabolic syndrome traits (central obesity, type 2 diabetes mellitus, hypertension, hyperlipidaemia and hypo-HDL cholesterolemia) and the edges correspond to experimentally supported associations. The size of the nodes corresponds to the nodes' degree.

syndrome. In addition, the lncRNA with the highest degree is H19, which is related to obesity, type 2 diabetes mellitus, and hypo-HDL cholesterolemia, and is reported to be involved in the pathogenesis of metabolic syndrome. Further and deeper analysis of the ncRNA2MetS data will yield more interesting results.

## DISCUSSION

Metabolic syndrome has become one of the most important diseases threatening human health worldwide. There is increasing evidence suggesting that metabolic syndrome is associated with abnormal expression of some ncRNAs, including miRNAs and lncRNAs. A database dedicated to metabolic syndrome-ncRNA association is helpful in studying the pathogenesis and treatments of metabolic syndrome. For this purpose, we developed ncRNA2MetS, a database containing almost all experimentally supported metabolic syndrome-ncRNA associations. Currently, ncRNA2MetS contains 1,068 validated associations between five metabolic syndrome traits and 627 ncRNAs (543 miRNAs and 84 lncRNAs) in four species.

In recent years, some researchers have developed several high-quality databases, such as BioM2MetDisease (*Xu et al., 2017*) and HMDD (*Huang et al., 2018*), to provide metabolic disease-miRNAs associations. Nevertheless, these databases are not dedicated to metabolic syndrome and do not cover all the metabolic syndrome traits. For example, BioM2MetDisease contains 2,681 entries of relationships between 524 miRNAs and 45

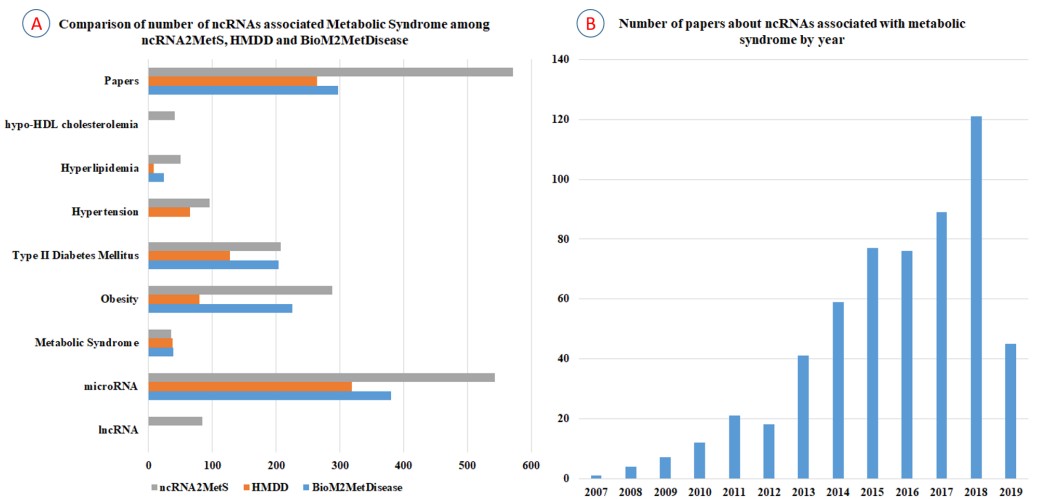

**Figure 5  Comparison of the number of ncRNAs associated with metabolic syndrome in different databases.** (A) Comparison of the number of ncRNAs associated with metabolic syndrome among BioM2MetDisease, HMDD and ncRNA2MetS. (B) Number of papers about ncRNAs associated with metabolic syndrome between 2007 and April 2019.

metabolic diseases across 14 species. This is a database with very rich storage content, but it is not specifically for metabolic syndrome. Though BioM2MetDisease contains a large number of miRNAs associated with obesity, type 2 diabetes mellitus and dyslipidemia, it lacks miRNA information related to hypertension and hypo-HDL cholesterolemia. Furthermore, although a large number of new miRNAs and lncRNAs related to metabolic syndrome have been identified and reported in the past two years, BioM2MetDisease has not been updated with the latest findings. HMDD (v3.0) is a database that curates experiment-supported evidence for human miRNA and disease associations. Currently, HMDD contains 32,281 miRNA-disease association entries which include 1,102 miRNA genes and 850 diseases. Similar to BioM2MetDisease, HMDD is not specifically for metabolic syndrome, and it does not contain miRNAs associated with hypo-HDL cholesterolemia. Furthermore, the number of miRNAs contained in HMDD is far less than ncRNA2MetS. To demonstrate the value of ncRNA2MetS, we comprehensively compared the amount of non-coding RNAs associated with various metabolic syndrome traits contained in BioM2MetDisease, HMDD and ncRNA2MetS. The results showed that ncRNA2MetS contained significantly more miRNAs associated with metabolic syndrome than both BioM2MetDisease and HMDD (Fig. 5A). More concretely, the number of miRNAs associated with obesity, type 2 diabetes mellitus, hypertension, hypertriglyceridemia, hypo-HDL cholesterolemia and metabolic syndrome were 225, 204, 0, 24, 0 and 39 in BioM2MetDisease; 80, 128, 65, 8, 0 and 38 in HMDD; and 288, 207, 96, 50, 41 and 36 in ncRNA2MetS, respectively. Finally, ncRNA2MetS provides not only miRNAs but also lncRNAs associated with metabolic syndrome, and covers the latest research findings up to April 2019.

The prevalence of metabolic syndrome, including central obesity, type 2 diabetes mellitus, hypertension, hypertriglyceridemia and hypo-HDL cholesterolemia, is growing globally, and the amount of studies on metabolic syndrome is also increasing rapidly. To illustrate the research trend of ncRNAs related to metabolic syndrome, we counted the number of articles about non-coding RNA studies related to metabolic syndrome between 2007 and April 2019 (Fig. 5B). Clearly, the amount of research about the association between ncRNAs and metabolic syndrome has increased rapidly in the past two years. Therefore, it will become a trend that more and more metabolic syndrome-ncRNA associations will be identified and validated in the future. We will keep track of the latest advances in the study of relationship between metabolic syndrome and non-coding RNA, and update ncRNA2MetS database regularly. In addition, we will focus on more types of non-coding RNA such as circular RNA, snoRNA, etc. and add associations between metabolic syndrome and these ncRNAs into ncRNA2MetS to increase its coverage. Furthermore, we will develop more powerful data analysis tools such as network visualization tool to help researchers better study the pathogenesis and treatment of metabolic syndrome in the future. In a word, we hope that ncRNA2MetS can be used as an effective tool for studying the mechanism of non-coding RNAs in metabolic syndrome.

## CONCLUSIONS

A growing number of studies have suggested that many non-coding RNAs, including miRNAs and lncRNA, are involved in metabolic syndrome and its traits. In this article, we introduced ncRNA2MetS, a user-friendly web-based tool developed for curating the association between metabolic syndrome and ncRNAs (miRNA and lncRNAs). ncRNA2MetS currently contains 1,068 associations between five metabolic syndrome traits and 627 ncRNAs (543 miRNAs and 84 lncRNAs) in four species. ncRNA2MetS has covered almost all relevant researches about the association between metabolic syndrome and ncRNAs between 2007 and 2019. It is expected that ncRNA2MetS will serve as a valuable data resource that will help researchers better study the pathogenesis and treatments of metabolic syndrome.

### Funding
This work was supported by the Harbin Science and Technology Innovation Talents Research Project (No. 2017RAQXJ027), Youth Innovative Talents Training Program for Universities of Heilongjiang Province (No. UNPYSCT-2018208) and China Scholarship Council. The funders had no role in study design, data collection and analysis, decision to publish, or preparation of the manuscript.

### Grant Disclosures
The following grant information was disclosed by the authors:
Harbin Science and Technology Innovation Talents Research Project: 2017RAQXJ027.

Youth Innovative Talents Training Program for Universities of Heilongjiang Province: UNPYSCT-2018208.
China Scholarship Council.

## Competing Interests

The authors declare there are no competing interests.

## Author Contributions

- Dengju Yao and Xiaojuan Zhan conceived and designed the experiments, performed the experiments, analyzed the data, contributed reagents/materials/analysis tools, prepared figures and/or tables, authored or reviewed drafts of the paper, approved the final draft, develop the website of the database.
- Xiaorong Zhan conceived and designed the experiments, performed the experiments, analyzed the data, contributed reagents/materials/analysis tools, authored or reviewed drafts of the paper, approved the final draft.
- Chee Keong Kwoh conceived and designed the experiments, contributed reagents/-materials/analysis tools, authored or reviewed drafts of the paper, approved the final draft.
- Yuezhongyi Sun performed the experiments, analyzed the data, contributed reagents/materials/analysis tools, authored or reviewed drafts of the paper, approved the final draft, develop the website of the database.

## Data Availability

The raw data are available as a Supplemental File.

## Supplemental Information

Supplemental information for this article can be found online at http://dx.doi.org/10.7717/peerj.7909#supplemental-information.

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
