# Peer review of "ncRNA2MetS: a manually curated database for non-coding RNAs associated with metabolic syndrome"

_PeerJ, doi:10.7717/peerj.7909_

## Round 0.1 · original submission · Major Revisions

All three reviewers recommended major revision. First reviewer is a biologist, specialist on metabolic syndrome. Two others are computer scientists. The manuscript has critics from both sides, on practical advantages for medicine and data presentation. I hope you'd benefit from these complementary remarks to improve the work. Despite the critics I encourage update the text and resubmit the manuscript to PeerJ in due time.

·

Basic reporting

The manuscript is written in clear and unambigous language. However, I think that the term "hypolipoproteinemia" mentioned as one of the five metabolic syndrome traits and used in the database is misleading, as it may refer to low levels of both Low-Density Lipoprotein cholesterol (LDL-c) and High-Density Lipoprotein cholesterol (HDL-c) which perform opposite functions. The word "hypolipoproteinemia" is not mentioned in the articles from which the associations were retrieved, nor is it commonly used in the literature as a trait associated with metabolic syndrome. Apparently, the authors meant low levels of HDL-c, so I would suggest to use the term "hypo-HDL cholesterolemia" or a broader term "dyslipoproteinemia".

Experimental design

no comment

Validity of the findings

The authors describe in detail the interface of the database. Obviously, it is well thought-out and convinient to use. Unfortunately, the website mentioned in the article is not available. I would like to explore it so that my review is complete.

Additional comments

The manuscript describes a manually curated database for noncoding RNAs associated with metabolic syndrome. This work is of great importance since metabolic syndrome has become a global threat to human health. The ncRNA2MetS database may become a valuable tool that will help researchers better study the pathogenesis of metabolic syndrome and associated diseases. In my opinion, the paper is suitable for publication in PeerJ after some revision and after the web interface is made available.

Some minor comments:

line 56: "including small non-coding RNAs (miRNAs)" As miRNA is an abbreviation for microRNA, I would suggest to rephrase to "including small non-coding RNAs, particularly microRNAs (miRNAs)"

line 106: "To meet demand" - "to meet this demand"

line 136: "DO identifier" - it is better to explain that DO refers to Desease Ontology

line 184: "the website will return all MetS-ncRNA associations meet the query" - "associations that meet the query"

line 186: "validated methods" - "validation methods"

line 219: "PARAL1" - maybe change to "Paral1"?

line 220: "The introduction to these associations in ncRNA2MetS show that" - "shows that"

line 225: "lincRNA" - "lncRNA"

line 241: "the relationship between metabolic and non-coding RNA" - "metabolic syndrome"

Figure 3: it is not intuitively clear why the ncRNAs in the central circle that are linked to multiple traits are of different color.

Figure 4: instead of "the comparison" and "a comparison" maybe use just "comparison"?
"non-coding RNAs associated metabolic syndrome" - "associated with metabolic syndrome"

Reviewer 2 ·

Basic reporting

In this manuscript, the authors developed a metabolic syndrome-associated non-coding RNA database (ncRNA2MetS) to curate the associations between metabolic syndrome and non-coding RNAs, and then constructed a user-friendly website so that users can access and download all data easily. While this work might provide useful resources for the future metabolic syndrome-associated non-coding RNA study, I have some concerns:

1. In the ncRNA2MetS website, the authors should provide more related information, such as the RNA sequences, explanation and/or external links of disease ontology ID and ICD-10 classification code.

2. In order to better study the correlation between these ncRNA genes and metabolic syndrome, the authors should integrate and display more omics data of these genes, such as these genes’ multi-tissue expression data (especially in the disease-related tissues), in the database.

3. The authors should provide more online tools for users to better explore the ncRNA2MetS database, such as the NCBI online BLAST service.

4. Since this manuscript is presenting a database, it will be good to include a diagram of the data model, eg. ER model, to show some technical details about the database design.

Experimental design

NA

Validity of the findings

NA

Reviewer 3 ·

Basic reporting

The authors developed a metabolic syndrome-associated non-coding RNA database (ncRNA2MetS) to curate the associations between metabolic syndrome and non-coding RNA. The ncRNA2MetS contains 1068 associations between five metabolic syndrome traits and 627 non-coding RNAs (543 miRNAs and 84 lncRNAs) in four species. They also developed a user-friendly website so that users can access and download all data easily. The ncRNA2MetS is a complete and high-quality data resource for exploring the role of non-coding RNA in the pathogenesis of metabolic syndrome and seeking new treatment options. The website is freely available at http://www.biomed-45 bigdata.com:50020/

Experimental design

The authors claim that there is no non-coding RNA database dedicated to metabolic syndrome. The authors need to comment if the database search engine can include other areas of disease that results from MetSyn such as gut bacteria dysbiosis, inflammatory bowel disease, cancer, etc.

Validity of the findings

The authors manually collected and curated MetS-miRNA (lncRNA) association from related articles in the PubMed database. They used ‘non-coding RNA’, ‘ncRNA’, ‘microRNA’, ‘miRNA’, ‘long non-coding RNA’, ‘lncRNA’ and each metabolic syndrome trait as search terms to search the PubMed database by Title/Abstract retrieval method. As a result, they acquired more than 3000 related articles published since 2007.

Additional comments

1. Metabolic syndrome is a cluster of the most dangerous heart attack risk factors .... Please define MetSyn properly.
2. miRNAs were shown to play critical roles in core processes of the insulin-related signaling pathway, carbohydrate and lipid metabolism, as well as adipocytokine signaling pathways. Up- or downregulation of certain miRNAs has been correlated with the development of insulin resistance and increased severity of T2DM. In particular, miR-7 was found to regulate pancreatic beta-cell function, differentiation and insulin secretion. Overexpression of miR-7 in mice has been associated with the development of T2DM due to impaired insulin secretion. Also, levels of miR-101, miR-375, miR-802 were significantly increased in T2DM patients while miR-143 and miR-223 levels were downregulated in obese individuals when compared to control groups. Constant exposure of pancreatic beta-cells to various metabolic stresses, including sweetener consumption, can shift the delicate balance between positive and negative regulatory miRNA, ultimately promoting pancreatic dysfunction and insulin resistance. Importantly, miRNAs can affect different bacterial species due to their capacity to enter the GI tract. This effect can partially explain the differences observed in miRNA regulatory effects.
3. The authors need to check the most recent articles on MetSyn and explain how the different conditions of disease elements may affect database profiling, additional parameters, and results.
4. What are the limitations of the database analyses?

---

## Round 0.2 · Minor Revisions

There are no critical remarks. All the comments are related to language editing. Please check it again, as suggested.

·

Basic reporting

The authors have taken into account the previous comments. In my opinion, no major revision is needed.

Experimental design

no comment

Validity of the findings

no comment

Additional comments

I see now that the web-site is generally available. I still cannot access it from my institution (which is strange), but it is available through my mobile Internet provider.

Some minor comments:

I have a little suggestion about the search page of the Web-interface. It would be great to make the fuzzy search more fuzzy (in the future). For example, the search for "mir 155" as microRNA type gives no results.

line 40: "specie involved" - "species involved"
line 70: "to develop database" - "to develop a database"
line 110: "various metabolic syndrome trait" - "traits"
lines 198, 199, 201: "validated methods" - "validation methods", the same for the Web-interface
line 256: "these three miRNAs plays an important role" - "play"

Reviewer 2 ·

Basic reporting

1. The figs are not easy to understand, and I can not find the detailed legends of their figs.
For example, in fig1, what is "other resources" ?
2. The authors need to find a native speaker to revise their writing.

Experimental design

NA

Validity of the findings

NA

Additional comments

NA

Reviewer 3 ·

Basic reporting

acceptable

Experimental design

acceptable

Validity of the findings

acceptable

Additional comments

none

---

## Round 0.3 · accepted · Accept

Thank you for the manuscript update. There are no remarks from the reviewers.

·

Basic reporting

no comment

Experimental design

no comment

Validity of the findings

no comment

Reviewer 2 ·

Basic reporting

no comment

Experimental design

no comment

Validity of the findings

no comment